# Reprogramming the Apoptosis–Autophagy Axis in Glioblastoma: The Central Role of the Bcl-2:Beclin-1 Complex and Survival Signalling Networks

**DOI:** 10.3390/cells15010053

**Published:** 2025-12-27

**Authors:** Monika Christoff, Amelia Szczepańska, Joanna Jakubowicz-Gil, Adrian Zając

**Affiliations:** 1Faculty of Biology and Biotechnology, Maria Curie-Skłodowska University, Akademicka 19, 20-033 Lublin, Poland; monikachristoff1@gmail.com (M.C.); amelia2003.30@wp.pl (A.S.); 2Department of Functional Anatomy and Cytobiology, Institute of Biological Sciences, Maria Curie-Sklodowska University, Akademicka 19, 20-033 Lublin, Poland; joanna.jakubowicz-gil@mail.umcs.pl

**Keywords:** glioblastoma multiforme, Bcl-2:beclin-1 complex, apoptosis, autophagy, programmed cell death axis, cancer cell, survival pathways, molecular switch

## Abstract

Glioblastoma multiforme (GBM) exhibits remarkable resistance to therapy, mainly due to its capacity to modulate regulated cell death pathways. Among these, apoptosis and autophagy are dynamically interconnected, determining cell fate under therapeutic stress. The interaction between beclin-1 and Bcl-2 proteins may represent a key molecular switch that controls whether glioma cells undergo survival or death. This review highlights the crucial role of the Bcl-2:beclin-1 complex in controlling apoptosis–autophagy axis in GBM, emphasising how survival signalling networks, including PI3K/AKT/mTOR, Ras/Raf/MEK/ERK, and PLCγ1/PKC pathways regulated by the TrkB receptor, modulate this balance. We summarise recent insights into how these pathways coordinate the shift between apoptosis and autophagy in glioma cells, contributing to drug resistance. Furthermore, we highlight how modulating this crosstalk can sensitise GBM to conventional and emerging therapies. Integrating new concepts of cell death reprogramming and systems-level signalling analysis, we propose that targeting the Bcl-2:beclin-1 complex and its upstream regulators could overcome the adaptive plasticity of glioblastoma multiforme and open new directions for combination treatment strategies.

## 1. Introduction

Gliomas are primary brain cancers categorised by cell origin. They comprise astrocytic tumours, ependymomas, and oligodendrogliomas. According to a WHO report, glioblastoma multiforme (GBM) is considered a grade IV brain cancer, and its heterogeneous malignancy occurs *de novo* or as a result of a secondary disease. GBM is characterised by high cellular and molecular heterogeneity, intensive proliferation, vascularisation, and infiltrating growth, resulting in a poor prognosis and disappointing cure rates. It correlates with exceptionally high mortality and inevitable recurrence, making it one of the most challenging intracranial malignancies with a median overall survival of only 16–18 months and a 5-year survival rate of less than 3% [1]. Despite tremendous advances in medicine and significant progress in research into the molecular biology and genetics of this type of cancer, the treatment of gliomas still faces many obstacles. Therefore, great emphasis is placed on gaining a more thorough understanding of the molecular mechanisms responsible for the high migratory potential and proliferation index of these cells, with a view to developing modern therapeutic strategies that eliminate glioblastoma cells through programmed cell death. Based on the current understanding of molecular mechanisms, apoptosis appears to be the preferred mechanism of cell death. However, some types of cancer frequently develop resistance to this process, including primary gliomas. Part of this resistance is based on increased transmission of intracellular pathways, such as PI3K/Akt/mTOR, Ras/Raf/MEK/ERK, and PLCγ1/PKC, which are regulated by TrkB receptor activity. Thus, inhibiting these pathways may decrease GBM’s intrinsic and drug-induced resistance and sensitise its cells to apoptosis. Conversely, the induction of alternative non-apoptotic cell death mechanisms may be essential for eliminating apoptosis-resistant GBM cells [2,3,4]. In cancer, autophagy has been shown to be an important anti-cancer mechanism in vivo, as the expression level of beclin-1 (a marker autophagy gene) is inversely correlated with the malignancy of brain tumours and directly correlated with survival. Furthermore, it has been demonstrated that GBM cells are more susceptible to agents that induce autophagy than apoptosis, such as temozolomide (TMZ), and autophagic structures have been observed in vivo in gliomas after treatment [5,6]. However, autophagy appears to have a dual role in cancer, as it has now been shown that autophagy also facilitates the survival of tumour cells in stressful conditions, such as hypoxic or nutrient-poor environments. This should be taken into account when designing new treatments. Interestingly, autophagy and apoptosis have also been shown to be interconnected by several molecular nodes of crosstalk, offering novel therapeutic opportunities and enabling the coordinated regulation of degradation by these pathways [7].

Therefore, a comprehensive understanding of the interconnectivity of autophagy and apoptosis is essential for developing effective cancer therapeutics. In this review, we focus on the multiple cellular and molecular mechanisms coordinating autophagy and apoptosis, with a particular emphasis on TrkB receptor pathways and the proteins involved in this crosstalk in cancer. While several recent reviews comprehensively describe autophagy-apoptosis interplay in GBM, few explicitly integrate the role of TrkB receptor signalling networks (PI3K/Akt/mTOR, Ras/Raf/MEK/ERK, and PLCγ1/PKC) as central modulators of this crosstalk. Our review highlights how dysregulation of these pathways directly influences the apoptosis–autophagy decision node. Although the existing literature mostly discusses the Bcl-2 family and beclin-1 in isolation, our manuscript systematically frames this interaction as a dynamic regulatory switch at the intersection of survival signalling and therapeutic reprogramming in GBM. We also emphasise a translational viewpoint that goes beyond descriptive summaries by proposing how targeting upstream signalling hubs that modulate the Bcl-2:beclin-1 axis may sensitise GBM cells to death pathways, supported by experimental and preclinical evidence.

## 2. Molecular Architecture of the Apoptosis–Autophagy Axis in Glioma Cells Correlated with Survival Pathways Signalling

The molecular architecture that regulates apoptosis and autophagy processes in glioma, especially GBM cells, is a highly adaptive surveillance network. It is a cell-death regulatory network, which interacts with multiple pro-survival signalling cascades that are frequently overexpressed during tumorigenesis. The apoptosis proceeds through two major pathways: the intrinsic or extrinsic mechanism of controlled cell death that eliminates the cancerous glial cells [8]. On the other hand, the autophagy is considered a context-dependent, dual-role process in GBM cells [9]. Glioblastoma cells develop therapy resistance through autophagy induction under hypoxia. Thus, autophagy’s role in glioblastoma multiforme is complex because it can support GBM growth, but excessive regulation of autophagy can inhibit proliferation and induce cancer cell death [10]. In addition, deficiencies in genes regulating autophagy, like beclin-1, the key mediator of autophagy and disturbances in the survival signalling pathways, are linked to glioblastoma development [11]. In the case of glioblastoma multiforme cells, mechanisms responsible for inducing programmed cell death, such as apoptosis, are blocked. At the same time, there is an overexpression of chaperone proteins that promote the survival of cancer cells [12]. Moreover, GBM is one of the most resistant tumours to apoptosis, partly due to the accumulation of mutations in mitochondrial DNA, which contributes to mitochondrial dysfunction leading to abnormal energy production and reactive oxygen species, associated with resistance to apoptosis [13]. As it was reported before in our works and others within the literature [9,14,15,16], apoptosis and autophagy are co-existing processes in glioma cells and may influence each other. Importantly, when cellular stress exceeds compensatory capacity, crosstalk between autophagic and apoptotic regulators determines whether the cell transitions into programmed death or shifts into an alternative survival state. The resulting plasticity not only shapes tumour progression but also poses a major obstacle for therapeutic intervention.

The malignant character of high-grade gliomas and their PCD-resistance is closely linked to the severe activation of the TrkB receptor and its downstream PI3K/Akt/mTOR, Ras/Raf/MEK/ERK, and PLCγ1/PKC pathways overexpression, which normally regulate neuronal and glial survival, proliferation, and differentiation but become dysregulated to sustain uncontrolled growth (Figure 1). While selected examples from our previous experimental studies are discussed where they directly inform mechanistic aspects of the apoptosis–autophagy crosstalk, the signalling pathways described below are supported by a broad body of independent experimental and clinical evidence.

Continuous signalling through these cascades induces excessive proliferation, accumulation of DNA mutations, and the progressive transformation of glial cells into malignant phenotypes. Within this network, the PI3K/Akt/mTOR pathway plays a crucial role. The RTK-dependent PI3K activation converts PIP2 to PIP3, enabling full Akt activation through PDK1 and mTOR-mediated phosphorylation, with the three Akt isoforms exerting distinct yet complementary roles in growth, metabolism, and neural development. Hyperphosphorylated Akt correlates with aggressive glioma biology, enhanced temozolomide and radiotherapy resistance, and suppression of apoptosis through inhibition of caspase-9 and stabilisation of anti-apoptotic Bcl-2. Simultaneously, Akt-mediated inactivation of GSK3β promotes β-catenin-driven survival programs, while mTORC1 suppresses autophagy, a process that may either protect glioma cells or shift toward a death-promoting role depending on context [17,18,19,20,21,22,23]. Based on our previous work [24], experimental inhibition of PI3K using LY294002 in T98G and MOGGCCM cells reduces PI3K/Akt/mTOR signalling, induces programmed cell death, and differentially activates apoptosis or autophagy between lines. When combined with TMZ, shifting toward apoptotic dominance markedly increases GBM cells’ sensitivity. Parallel to PI3K/Akt/mTOR, the Ras/Raf/MEK/ERK cascade also drives proliferation and apoptosis resistance in glioma cells [25,26,27,28,29,30]. Ras activation promotes pathway phosphorylation events that culminate in transcriptional programs regulating proliferation, invasion, and survival, while Raf additionally supports resistance by engaging IAP family proteins that inhibit caspases. Common EGFR alterations and Ras/Raf/MEK/ERK pathway overexpression contribute to glioma therapy resistance [31,32]. Our experimental studies show that combining sorafenib with temozolomide [33] or LY294002 or furanocoumarins [34] enhances autophagy and apoptosis in T98G cells, while dihydroartemisinin suppresses MEK/ERK phosphorylation and Bcl-2/Mcl-1 in different glioma models [35]. Further complexity of GBM cells’ resistance is also correlated with overexpression of PLCγ1/PKC downstream of TrkB [36,37,38]. While PLCγ1 normally participates in neuronal and astrocytic plasticity, gliomas co-opt this signalling architecture to promote proliferation, motility, and survival, and elevated *PLCG1* expression correlates with aggressive disease in multiple transcriptomic analyses. Functional inhibition of PLCγ1 reduces glioma growth and migration, and TrkB-dependent PLCγ1 and Ras/ERK activation support resistance to apoptosis. Some of our previous works showed that blockade of these pathways may affect the specific interactions between anti-apoptotic Bcl-2 and autophagy regulator beclin-1, leading to their complex formation and promoting cell death. Although direct glioma-specific evidence linking PLCγ1/PKC-mediated phosphorylation to disruption of the Bcl-2:beclin-1 complex is limited, existing studies demonstrate that PLCγ1-driven Ca^2+^ flux modulates autophagy-apoptosis balance by influencing beclin-1 availability at the ER membrane [36,39,40,41,42,43]. Overall, the overexpression and constitutive activation of these interconnected pathways shape the molecular architecture that enables glioma cells to evade apoptosis, sustain growth, and resist therapy.

## 3. The Bcl-2:Beclin-1 Complex as a Molecular Switch

Programmed cell death pathways often coexist within cells, as well as in gliomas, and exhibit mutual regulation through shared molecular mechanisms. Interactions between key regulators of apoptosis (PCD-I) and autophagy (PCD-II) illustrate this interdependence. For instance, Atg proteins involved in autophagy can modulate mitochondrial membrane potential, thereby influencing apoptotic signalling. Conversely, apoptosis-related proteins may inhibit specific Atgs, shifting the balance toward autophagy [44,45,46]. A well-known example, described in malignant glioma cells by our team before [14], of this crosstalk is the interaction between the anti-apoptotic protein Bcl-2 and the autophagy regulator beclin-1.

### 3.1. Structural and Biochemical Aspects

Members of the BCL-2 family are central regulators of mitochondrial outer membrane integrity and apoptosis. The anti-apoptotic subset, including Bcl-2, BCL-XL, MCL-1, BCL-W, and BFL-1/A1, preserves cell viability by sequestering pro-apoptotic BH3-only proteins (such as BID, BIM, or PUMA) and by directly inhibiting the pore-forming effectors BAX and BAK. Structurally, these pro-survival proteins share a conserved helical bundle fold composed of BH1-BH4 domains, with a hydrophobic groove formed mainly by BH1-BH3 segments that accommodates the amphipathic BH3 helix of their binding partners (Figure 2a). The occupancy of this groove determines whether the cell remains in a survival state or progresses toward mitochondrial outer membrane permeabilization and apoptosis [47,48,49,50,51].

Beyond their canonical anti-apoptotic function, BCL-2 family proteins also influence autophagy, a process of lysosomal degradation critical for cellular homeostasis and stress adaptation. This crosstalk is mediated in large part through direct physical interaction between Bcl-2/BCL-XL and a core autophagy regulator beclin-1 (*BECN1*). It is an obligate regulatory subunit of the class III phosphatidylinositol 3-kinase (PI3K-III/VPS34) core complex that nucleates nascent autophagosomal membranes and recruits additional cofactors that determine complex specificity (e.g., ATG14L for the autophagy-initiating PI3KC3-C1). This architectural role of beclin-1 within the Vps34–Vps15–beclin-1 core is well established in biochemical and cryo/structural analyses [52,53,54,55,56].

Structurally, beclin-1 consists of three major segments that guide its conformational flexibility and functional interactions (Figure 2b). The C-terminal BARA domain (residues ~265–450) forms a compact, β–α–repeated autophagy-related fold, responsible for membrane binding and complex stabilisation. The BARA domain interacts with lipids and other components of the autophagic membrane, anchoring beclin-1 to sites of autophagosome nucleation. The central coiled-coil domain (residues ~150–265) mediates beclin-1 homodimerization and heterodimerization with other autophagy regulators, such as ATG14L or UVRAG. These interactions determine the functional identity of the PI3K-III complexes and their subcellular targeting. Structural studies and AlphaFold predictions show that this region adopts an extended α-helical conformation, promoting stable intermolecular pairing essential for complex assembly [57,58,59,60,61]. The N-terminal portion (residues ~1–150) is predominantly intrinsically disordered, which allows it to act as a flexible regulatory platform for post-translational modifications and partner binding. Within this region resides a BH3-like motif (~residues 105–130), a short amphipathic α-helix that engages anti-apoptotic BCL-2 family proteins such as Bcl-2 and BCL-XL. This helix docks into the hydrophobic groove of the BCL-2 proteins in a manner analogous to canonical BH3-only ligands (e.g., BIM, BID). By this interaction, beclin-1 is sequestered by Bcl-2/BCL-XL at intracellular membranes, including the endoplasmic reticulum (ER) and mitochondria, thereby suppressing autophagy initiation under nutrient-rich conditions [49,62,63,64,65].

**Figure 2 cells-15-00053-f002:**
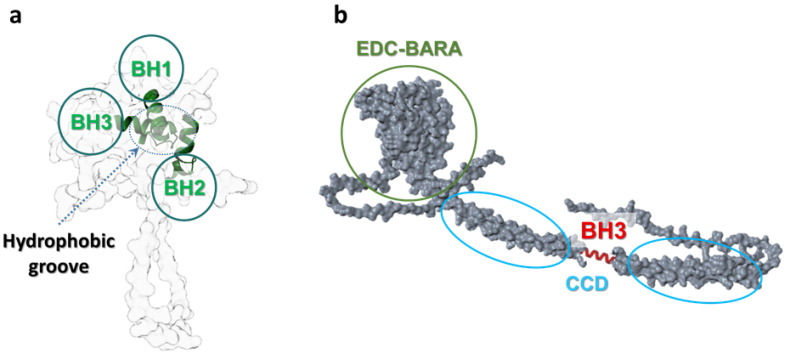
Scheme of structures and domains’ conformations of Bcl-2 (**a**) and beclin-1 (**b**) proteins. BH1-3 (green)—Bcl-2 homology domains 1,2, and 3 of Bcl-2; BH3 (red)—Bcl-2 homology 3-like motif of beclin-1; CCD—coiled-coil domain; EDC—evolutionarily conserved domain; BARA—β–α–repeated autophagy-related fold. This domain-level view explains how Bcl-2 family proteins recognise BH3-like helices and why beclin-1 can be sequestered by Bcl-2 to restrain autophagy initiation. UniProt IDs for created conformations: Bcl-2: AF-P10415-F1, beclin-1: AF-Q6ZNE5-F1. Molecular graphics and analyses performed with UCSF ChimeraX [66].

When Bcl-2 (or BCL-XL) binds beclin-1 through the hydrophobic groove and BH3-like motif domain interaction (Figure 3), it sterically blocks its association with activating PI3K-III cofactors (ATG14L, VPS34, VPS15), thus suppressing autophagy initiation. Conversely, disruption of the Bcl-2:beclin-1 complex allows beclin-1 to form active PI3K complexes and promote autophagy. This binary, reversible binding underlies the idea of the Bcl-2:beclin-1 complex acting as a molecular “toggle switch” between survival and autophagy-permissive states [53,54,67,68].

### 3.2. Post-Translational Regulation

Beclin-1 function and its interaction with anti-apoptotic Bcl-2 proteins are tightly controlled by multiple post-translational modifications (PTMs). These PTMs act as rapid, reversible switches that return beclin-1 availability for PI3K-III complex formation, alter protein stability, or convert it into pro-death fragments. Below, we summarise the major PTM classes, the best-characterised sites and/or enzymes, and the functional outcomes relevant to cancer and glioma biology [69,70].

Phosphorylation is the most well-studied post-translational modification that may affect the Bcl-2:beclin-1 axis. Several kinases phosphorylate residues in either beclin-1 or Bcl-2 weaken their interaction and thereby promote autophagy such as DAPK/DAPK2. Among the kinases known to regulate the beclin-1and Bcl-2 interaction, the death-associated protein kinases (DAPKs) provide a particularly clear molecular example of how stress signals are translated into autophagy activation. The founding member of this family, DAPK1, is a Ca^2+^/calmodulin-regulated serine/threonine kinase, first linked to cell-death pathways and later recognised as an important modulator of autophagy [71,72,73]. Using biochemical and cellular approaches, Zalckvar et al. [71] identified beclin-1 as a direct substrate of DAPK1, pinpointing threonine 119 within its BH3-like domain as the principal phosphorylation site. This site occupies a strategic position within the amphipathic α-helix that mediates beclin-1’s binding to the hydrophobic groove of anti-apoptotic Bcl-2. In the unmodified state, Thr119 forms hydrophobic and hydrogen-bond contacts that stabilise this inhibitory complex. The addition of a negatively charged phosphate group by DAPK1 disrupts these interactions, producing a marked decrease in beclin-1’s affinity for Bcl-2. These biochemical observations were supported by structural modelling of the beclin-1-BH3-helix in complex with BCL-xL, which demonstrated that phosphorylation at Thr119 would introduce electrostatic repulsion and steric hindrance at the protein–protein interface [58,71,72,74]. Thus, phosphorylation at this single residue acts as a trigger that releases beclin-1 from its inhibitory sequestration, allowing it to join the class III PI3K (VPS34) complex and promote autophagosome nucleation. The 5’ adenosine monophosphate-activated protein kinase’s regulation of beclin-1 provides another key link between metabolic stress and the activation of autophagy, with a potential effect on the Bcl-2:beclin-1 interactions. When cellular energy levels fall, AMPK can directly modify beclin-1 at several residues in its N-terminal region, particularly Ser90, Ser93, and Ser96. These phosphorylation events help shift the protein toward a more active state, strengthening its ability to support VPS34 complex formation and thereby promoting autophagosome initiation. AMPK also influences beclin-1 indirectly through DAPK2, which can phosphorylate it at complementary sites and amplify the autophagic response in settings where calcium-dependent stress signals are present. Through this combination of direct phosphorylation and kinase crosstalk, AMPK turns beclin-1 to help cells adjust their autophagic machinery to the energetic landscape. Emerging work even suggests that this regulatory cluster may shape how this protein participates in processes such as ferroptosis, pointing to a broader role for AMPK-dependent beclin-1 modification in managing cellular stress outcomes [73,75,76,77].

Ubiquitination adds another important layer of control to beclin-1, with the type of ubiquitin chain determining whether the protein is stabilised or targeted for destruction. Several studies have shown that K63-linked ubiquitination, mostly mediated by TRAF6, supports autophagy by stabilising beclin-1 or by strengthening its assembly within the PI3K-III complex [78,79]. In contrast, K48-linked ubiquitin chains direct the protein toward proteasomal degradation, thereby lowering its availability for autophagic signalling and dampening autophagy over longer time scales. This balance is shaped by the interplay between different E3 ligases and deubiquitinases. Because these ubiquitin modifications affect beclin-1 stability, they serve as a slower, more sustained regulatory mode that complements rapid phosphorylation-based switches [69,79,80,81]. The activity of the beclin-1 protein may also be shaped by acetylation, which has emerged as an influential regulator of autophagosome maturation. Acetyltransferases such as p300 can acetylate this protein or its complex with VPS34 and inhibit autophagic progression, while the deacetylase SIRT1 can reverse these modifications. Beyond these reversible modifications, beclin-1 can also be irreversibly remodelled by proteolytic cleavage. Caspase-3 and related caspases cut it at defined sites, eliminating its pro-autophagic function and producing fragments capable of relocating to mitochondria to amplify apoptotic signalling. This cleavage acts as a feed-forward mechanism, ensuring that apoptosis has been triggered. Importantly, these post-translational modifications do not operate in isolation because, as mentioned before, phosphorylation can prime beclin-1 for downstream ubiquitination, AMPK-dependent pathways intersect with DAPK family kinases, and acetylation states often reflect the activity of broader nutrient-sensing networks [82,83,84,85].

At the same time, the Bcl-2 may also undergo the PTMs regulations with the consequences that result in interferences between it and beclin-1. The regulation of those interactions by c-Jun N-terminal kinases 1 has become an example of how PTM shapes autophagy responses. Under nutrient-poor conditions, JNK1 is activated and targets several residues in the flexible N-terminal loop of Bcl-2, most notably Thr69, Ser70, and Thr87. Phosphorylation at these sites reduces its ability to bind the BH3-like region of beclin-1, which normally keeps it in an inactive, sequestered state at intracellular membranes. When this inhibitory interaction is disrupted, beclin-1 is free to join VPS34-containing PI3K-III complexes that initiate autophagosome formation. Mutational studies support this switch-like behaviour. Phosphomimetic Bcl-2 mutants release beclin-1 and enhance autophagy, whereas phosphorylation-resistant mutants maintain the inhibitory complex. This JNK1-driven phosphorylation not only enables beclin-1-dependent autophagy during stress but can also influence how Bcl-2 interacts with other BH3-only proteins, underscoring how a single regulatory loop helps coordinate cell survival pathways during metabolic challenge [86,87,88,89,90].

### 3.3. Functional Consequences in Glioma

Glioblastoma multiforme (GBM) and other high-grade gliomas exploit the apoptosis-autophagy interface for adaptive survival and therapeutic resistance. Several glioma-focused experimental and clinical studies implicate beclin-1 expression and Bcl-2 family regulation in GBM biology so far [91,92,93].

Given the pronounced molecular heterogeneity of GBM, the functional output of beclin-1-dependent autophagy and its coupling to apoptosis is likely to differ across clinically relevant tumour states rather than representing a uniform programme. Transcriptomic subtype frameworks (proneural, classical, mesenchymal) are linked to distinct upstream drivers (e.g., PDGFRA/IDH1 in proneural, EGFR in classical, NF1-associated programs in mesenchymal) that converge on PI3K/Akt/mTOR and MAPK signalling and can thereby shift the balance of Bcl-2 family priming and beclin-1 availability [94]. In parallel, the G-CIMP/IDH-mutant context-enriched within proneural tumours and associated with improved outcome is what supports the need to interpret autophagy-marker prognostic studies in a subtype-aware manner, as baseline metabolism and stress responses differ substantially from IDH-wild type GBM [95,96]. Treatment-linked heterogeneity further intersects with this axis: MGMT promoter methylation is a major determinant of temozolomide benefit and thus shapes therapeutic selection pressure for stress-adaptive autophagy during TMZ exposure, including contexts where TMZ induces autophagy and engages beclin-1-linked regulatory nodes [97]. Finally, mesenchymal-state GBM, which is frequently therapy resistant and may exhibit a higher anti-apoptotic threshold in part through differential Bcl-2 family programs (e.g., enrichment of *BCL2A1* reported in mesenchymal GBM and IDH-wildtype settings), provides a plausible mechanism by which the Bcl-2:beclin-1 checkpoint could be biased toward autophagy-permissive survival rather than apoptosis [98]. Collectively, these observations argue that the prognostic and therapeutic meaning of “high autophagy/beclin-1” versus “low autophagy/beclin-1” should be evaluated with explicit consideration of subtype (IDH/G-CIMP; proneural vs. mesenchymal) and treatment context (MGMT/TMZ exposure), rather than as a single unified GBM biomarker.

Experimental glioma dedicated work has reproduced these molecular relationships and shown functional consequences relevant to therapy: overexpression of beclin-1 in U87 GBM cells increases autophagic flux and, in some contexts, augments apoptosis, whereas beclin-1 knockdown reduces autophagy and alters treatment responses, indicating context-dependent pro-survival versus pro-death roles. At the level of patient material, immunohistochemical and biochemical studies of high-grade gliomas have reported heterogeneous results. Several series found higher cytoplasmic beclin-1 (and correlated LC3B-II) associated with increased autophagy markers and, in some cohorts, longer survival or better response to therapy (notably within MGMT-methylated subgroups), while other studies show no simple unidirectional relationship between beclin-1 levels and outcome. Finally, the Bcl-2 family was, more broadly, implicated in GBM therapeutic resistance because its members simultaneously regulate apoptotic threshold and autophagy. Upregulation of anti-apoptotic Bcl-2/BCL-xL can suppress apoptosis and autophagy, shifting glioma cells toward survival mechanisms under cytotoxic stress. Experimental disruption of Bcl-2 function modifies both autophagy and sensitivity to chemo-radiation procedures in preclinical models. These mechanistic and translational data together explain why interventions that selectively perturb the Bcl-2:beclin-1 axis (for example, phosphorylation modulators or BH3-mimetics that alter specific protein–protein interactions) are being explored as strategies to tip the balance toward glioma cell death. However, at the same time, why simple readouts of beclin-1 expression alone provide variable prognostic signals across studies and tumour subsets [14,91,92,99,100,101,102].

Importantly, clinical and experimental literature support conflicting associations between beclin-1-linked autophagy and GBM outcome [101]. Several studies suggest that reduced beclin-1/autophagy markers may reflect impaired autophagic pathways (including autophagic cell death) and can associate with aggressive disease biology in subsets of glioma cohorts. Conversely, other patient-series and marker-based analyses report that high LC3 and/or beclin-1 (often alongside p62/SQSTM1) correlates with shorter progression-free or overall survival, consistent with an interpretation that active autophagy can function as a stress-adaptive, cytoprotective programme supporting tumour maintenance [103,104,105]. These discrepancies likely reflect biological context (treatment exposure, hypoxia, stem-like populations), tumour heterogeneity, and methodological limitations of static autophagy markers that do not directly report autophagic flux. In therapy settings, multiple independent studies show that stress-induced autophagy can support resistance, including hypoxia-associated radioresistance and TMZ contexts where autophagy inhibition or beclin-1 suppression can sensitise GBM models to treatment-induced cell death [106,107].

## 4. Therapeutic Reprogramming of the Apoptosis–Autophagy Axis

### 4.1. Concept of Reprogramming

The ability of glioblastoma multiforme (GBM) to dynamically modify controlled cell death pathways in response to therapeutic stress is essential to its survival. The shifting balance between autophagy, which often acts as a pro-survival mechanism, and apoptosis, which triggers cells toward permanent death, is a characteristic of this elasticity. Because these two processes are tightly correlated and because their crosstalk is heavily shaped by the Bcl-2:beclin-1 complex, therapeutic reprogramming aims to reshape this balance in a direction that disadvantages tumour survival. The knowledge that the apoptosis–autophagy axis is a process controlled by signalling intensity, metabolic context, and stress duration rather than a binary switch lies at the core of this idea. The Bcl-2:beclin-1 complex functions as a crucial molecular checkpoint: when the complex is weakened or disrupted, cells may start autophagy or become more vulnerable to apoptosis, depending on mitochondrial priming and upstream signalling cues. When the complex is tightly associated, autophagy is suppressed, and Bcl-2 maintains its anti-apoptotic capacity [85,108,109,110].

Therefore, therapeutic reprogramming aims to modify the regulatory nodes that determine how GBM cells interpret stress, particularly points influenced by PI3K/Akt/mTOR, Ras/Raf/MEK/ERK, and PLCγ1/PKC activity, rather than merely “block autophagy” or “activate apoptosis” [14,36,111]. Whether the beclin-1-dependent autophagy machinery promotes survival or moves toward cell death is ultimately determined by these signalling networks’ modulation of phosphorylation states, protein-protein interactions, and organelle dynamics. By targeting these upstream regulators, reprogramming strategies try to overcome the tumour’s intrinsic stress adaptation, reduce compensatory resistance mechanisms, and direct GBM cells toward more vulnerable states.

### 4.2. Strategies Based on Experimental Evidence

Because the survival capacity of GBM is deeply correlated in the signalling pathways described above, many therapeutic approaches now may aim to rewire the regulatory context of the Bcl-2:beclin-1 complex or to modulate the consequences of its interactions. Several experimentally validated strategies have emerged from this integrated perspective. A major approach focuses on Bcl-2’s inhibitory impact on beclin-1, so changing autophagy away from a protective mechanism and toward a state that cooperates with apoptotic signals. Different strategies are known within the literature, including our previous works [14,36]. For example, newly described BH3-mimetic molecules, BAU-243 and ABT737, bind Bcl-2 with high affinity and disrupt the Bcl-2:beclin-1 complex in GBM cells, leading predominantly to autophagic cell death rather than apoptosis and importantly, they reduced tumour growth in an in vitro and in vivo glioblastoma model [91,112]. In contrast, overexpression of beclin-1 in GBM cell lines has been reported to boost autophagic flow and, under specific conditions, accelerate apoptosis, most likely by releasing pro-apoptotic signals such as cytochrome c and activating caspases-3 and 9 [99,113]. As it was mentioned before, we had discovered that under specific conditions of blocking the intracellular survival pathways, the complex of Bcl-2 and beclin-1 proteins was a specific trigger for apoptotic dominant glioma cell elimination. The effective route was to inhibit the survival signalling networks that regulate this molecular switch. Inhibitors of the PI3K/Akt/mTOR pathway, for example, can reduce autophagy suppression and sensitize GBM cells to death triggers. Indeed, combined inhibition of PI3K and Raf (which also affects the Ras/Raf/MEK/ERK pathway) was shown to tilt the balance toward apoptosis over autophagy in glioma cells [14,36]. Moreover, combining reprogramming strategies with conventional therapies, such as chemotherapy or radiotherapy, holds special promise. Since such treatments typically impose metabolic or genotoxic stress that may activate protective autophagy, concurrently disrupting Bcl-2:beclin-1 interactions or inhibiting survival pathways may prevent autophagy-mediated rescue and allow apoptotic or autophagic cell death to proceed [114,115].

### 4.3. Biomarker and Translational Implications

Identifying biomarkers that represent the status of the apoptosis–autophagy axis is essential for translation because it reacts dynamically to stress and therapeutic intervention [116]. Several categories of markers associated with the Bcl-2:beclin-1 regulatory hub have gained attention in translational studies and may assist in patient stratification [117]. Because GBM is molecularly heterogeneous and treated under distinct therapeutic contexts, the translational meaning of Bcl-2:beclin-1–related markers (beclin-1, Bcl-2 family expression, LC3/p62, and phospho-patterns) is best interpreted in a subtype-aware framework. Established axes, such as MGMT promoter methylation and IDH mutation status, strongly influence therapy response and metabolic stress states, while transcriptional subtype (proneural/classical/mesenchymal) reflects dominant upstream signalling programs that converge on PI3K/Akt/mTOR and MAPK pathways. To make this stratification explicit, we summarize clinically actionable subtype and treatment contexts that may modify interpretation of the apoptosis–autophagy axis and inform patient selection for reprogramming strategies (Table 1).

Among these, the expression levels of beclin-1 and Bcl-2 are particularly informative, as their relative abundance and the stoichiometry of their interaction can indicate whether a tumour is more likely to rely on autophagy or apoptosis during therapy [118]. Equally important are the phosphorylation states of both proteins. Since signalling pathways such as Akt and ERK directly modify Bcl-2 and beclin-1, the resulting phospho-patterns may predict the tumour’s responsiveness to upstream kinase inhibitors [70,71]. For instance, elevated PI3K or Raf signalling frequently correlates with strong autophagy-mediated resistance, whereas changes in PLCγ1 activity may influence Ca^2+^-dependent control of autophagy. Together, these markers can be incorporated into molecular profiles that help determine whether a tumour will benefit more from autophagy inhibition, modulation of Bcl-2 family interactions, kinase-targeted therapies, or combinations thereof [119,120].

However, several limitations complicate their clinical application. Autophagy is inherently dynamic, and static measurements of markers such as LC3 or beclin-1 may fail to capture autophagic flux or functional state. Additionally, GBM exhibits profound inter- and intra-tumoural heterogeneity, making it difficult to rely on single biomarkers for accurate predictions [121]. Redundant survival pathways may further necessitate multi-parameter panels rather than isolated molecular indicators.

For these reasons, future clinical implementation will likely depend on integrated approaches that combine molecular profiling of the Bcl-2:beclin-1 regulatory network with assessments of PI3K/Akt/mTOR, Ras/Raf/MEK/ERK, and PLCγ1/PKC pathway activity, as well as functional assays that evaluate autophagy and apoptotic susceptibility. Such multimodal strategies may ultimately help identify tumours with high “reprogrammability,” in which shifting the apoptosis–autophagy balance could meaningfully enhance therapeutic response.

### 4.4. Barriers to Clinical Translation

Despite strong preclinical rationale, direct clinical implementation of strategies targeting the Bcl-2:beclin-1 checkpoint in glioblastoma multiforme has been constrained by several practical barriers. First, on-target toxicities limit systemic dosing of many Bcl-2 family inhibitors. Dual Bcl-2/BCL-XL antagonists such as navitoclax (ABT-263) are associated with dose-limiting thrombocytopenia, reflecting the dependence of platelets on BCL-XL for survival, which narrows the therapeutic window and complicates combination regimens [122,123,124]. Second, the blood-brain barrier’s (BBB) pharmacology remains a central challenge because even when compounds show systemic activity, central nervous system penetration and (more importantly) intratumoural exposure can be limited by the BBB and efflux transport. While the other antagonist ABT-199 (venetoclax) has been measurable in cerebrospinal fluid in hematologic CNS, such observations do not necessarily translate into effective GBM tumour delivery or durable target engagement within the brain microenvironment [125,126,127]. Third, glioblastoma multiforme exhibits substantial route redundancy and adaptive rewiring across both cell-death regulators and upstream survival networks. Inhibition of a single anti-apoptotic protein can be bypassed by compensatory reliance on other Bcl-2 family members (e.g., MCL-1) and by reactivation of PI3K/MAPK stress-response signalling that reshapes autophagy and apoptotic priming, arguing for rational combinations rather than monotherapy [93,126]. Finally, experience with clinically used autophagy inhibitors, such as chloroquine or hydroxychloroquine, combined with standard GBM therapy illustrates both the feasibility and limitations of axis modulation, including toxicity constraints and the difficulty of demonstrating clear benefit without robust CNS pharmacodynamic readouts [128,129]. Collectively, these issues suggest that future translation will likely require BBB-optimised or brain-targeted compounds, improved pharmacodynamic biomarkers of complex disruption/flux, and subtype-/state-guided combination strategies that account for redundancy in death-regulatory networks.

## 5. Conclusions

Glioblastoma remains one of the most difficult to treat human malignancies, due to its potential to dynamically remodel cell-death systems and activate adaptive survival pathways. In glioma cells, apoptosis and autophagy are integrated into a stress-responsive regulatory network defined by oncogenic PI3K/Akt/mTOR, Ras/Raf/MEK/ERK, and PLCγ1/PKC signalling. The Bcl-2:beclin-1 complex is important to this interaction, acting as a molecular switch that determines whether GBM cells participate in survival-promoting autophagy or undergo apoptosis. Glioma cells may avoid apoptosis, exploit autophagy, and resist treatment due to aberrant TrkB downstream signalling and altered BCL-2 family activity. Reprogramming this axis via kinase inhibitors, BH3-mimetics targeting Bcl-2:beclin-1, or regulation of crucial post-translational events seems to offer a viable way to push GBM cells into irreversible death and enhance responses to current therapies. In summary, these results suggest a change in the way we think about the therapy of malignant gliomas. Instead of focusing on just one death pathway or protein, effective treatment may require changing the way cancer cells make decisions by changing the nodes that control whether stress leads to survival, autophagy, or death.

Key unresolved questions remain before apoptosis–autophagy reprogramming can be translated into durable GBM therapies. First, it remains unclear how the functional output of the Bcl-2:beclin-1 checkpoint varies across GBM molecular contexts (e.g., IDH status, MGMT methylation, and mesenchymal vs. proneural states) and during treatment-induced evolution. Second, future studies should move beyond static autophagy markers and instead incorporate direct assessments of autophagic flux and, where feasible, measurements of Bcl-2:beclin-1 complex dynamics in patient-derived models. Third, the extent to which pathway redundancy (PI3K/MAPK rewiring and compensation among Bcl-2 family members) limits single-node interventions needs systematic mapping to design rational combinations that avoid adaptive escape. Finally, clinical translation will require improved BBB-optimised delivery strategies, better pharmacodynamic readouts of on-target engagement in the brain, and validated biomarker panels that can identify tumours with high “reprogrammability.” Addressing these priorities should clarify when and how reprogramming the apoptosis–autophagy axis can be leveraged to overcome GBM therapeutic resistance.

## Figures and Tables

**Figure 1 cells-15-00053-f001:**
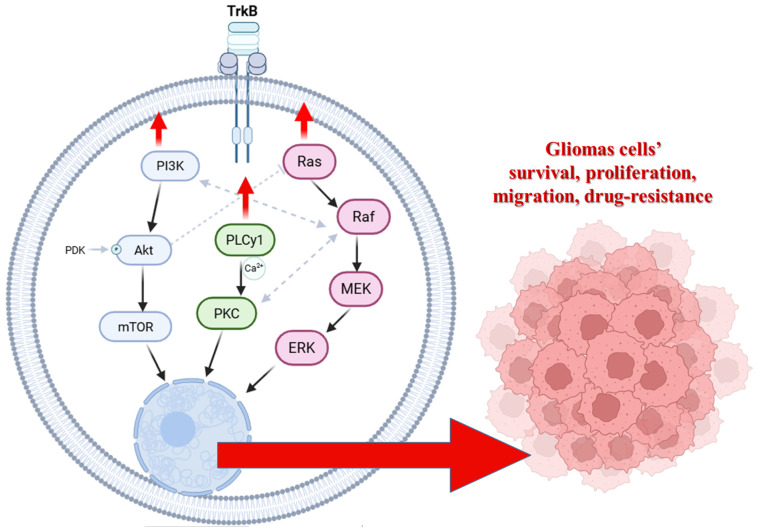
Visualisation of intracellular survival signalling in GBM cells via TrkB receptor and a downstream overexpression (red upper arrows) of regulated pathways: PI3K/Akt/mTOR, Ras/Raf/MEK/ERK and PLCγ1/PKC, and their interplay, leads to glioma cells survival, uncontrolled proliferation, migration, and drug resistance. Overall, the figure highlights how sustained TrkB-driven signalling amplifies survival and therapy resistance by simultaneously suppressing apoptotic execution and shaping autophagy responses. Created with BioRender.com.

**Figure 3 cells-15-00053-f003:**
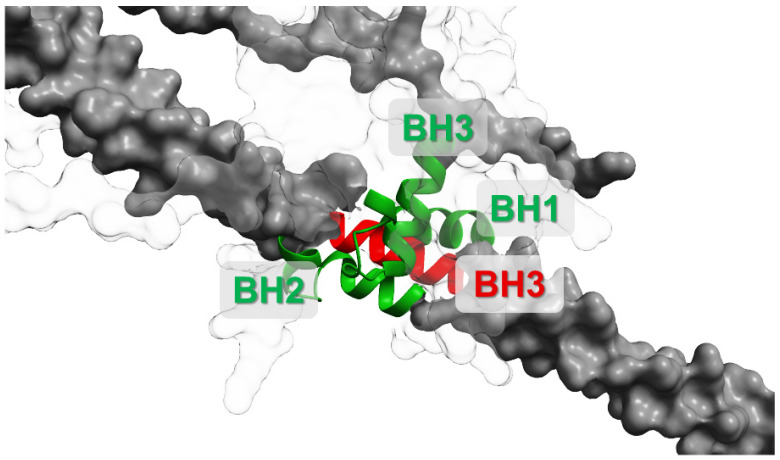
Scheme of Bcl-2:beclin-1 complex formations associated with Bcl-2 (white) and beclin-1 (grey) domains’ interactions. BH1-3 (green)—Bcl-2 homology domains; BH3 (red)—Bcl-2 homology 3-like motif. This interaction interface represents a mechanistic checkpoint: disrupting the complex can release beclin-1 to promote autophagy and can also alter apoptotic priming depending on upstream signalling and stress intensity. UniProt IDs for created conformations: Bcl-2—AF-P10415-F1, beclin-1—AF-Q6ZNE5-F1. Molecular graphics and analyses performed with UCSF ChimeraX [66].

**Table 1 cells-15-00053-t001:** Subtype- and treatment-aware interpretation of biomarkers linked to the Bcl-2:beclin-1 axis in glioblastoma.

GBM Context/ Biomarker Axis	Clinical Significance	Implication for Interpreting Bcl-2:Beclin-1/ Autophagy Markers	Translational Use
MGMT promoter methylation	Predicts benefit from TMZ in newly diagnosed GBM	TMZ selection pressure differs by MGMT status; TMZ can induce autophagy and engage beclin-1-linked regulation, so “autophagy-high” may reflect adaptive resistance, particularly under treatment	Consider MGMT status when evaluating autophagy inhibition/Bcl-2 targeting combinations with TMZ
IDH mutation/ G-CIMP	Defines a biologically distinct subgroup with improved outcome	Baseline metabolism and stress response differ vs. IDH-WT; autophagy-marker prognostic direction may not generalise across IDH states	Report autophagy/apoptosis readouts stratified by IDH state to avoid contradictory pooled conclusions
Transcriptional subtype (Proneural/ Classical/ Mesenchymal)	Captures dominant signalling programs	Upstream pathway dominance (EGFR/PI3K vs. NF1/mesenchymal inflammatory programs) likely shifts apoptotic priming and beclin-1 sequestration/availability	Use subtype calls to contextualise whether Bcl-2:beclin-1 disruption is predicted to push cells toward apoptosis vs cytoprotective autophagy
Mesenchymal-associated anti-apoptotic program	Associated with mesenchymal subtype/IDH-WT and poorer outcomes	Suggests higher anti-apoptotic buffering; Bcl-2-family dominance may alter the functional output of autophagy induction	Candidate marker for prioritising BH3-mimetic/Bcl-2-family targeting alongside autophagy modulation
Autophagy markers vs. flux context	Static IHC often fails to reflect flux; treatment state/hypoxia confounds	Helps explain why “high beclin-1/LC3” can associate with either favourable or unfavourable outcomes, depending on subtype and therapy exposure	Pair static markers with treatment/subtype annotation (MGMT/IDH/subtype) and, when possible, flux-oriented interpretation in translational studies

## Data Availability

No new data were generated.

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
