# Peer review of "Reprogramming the Apoptosis–Autophagy Axis in Glioblastoma: The Central Role of the Bcl-2:Beclin-1 Complex and Survival Signalling Networks"

_cells, 2025, doi:10.3390/cells15010053_

Round 1
Reviewer 1 Report
Comments and Suggestions for Authors
The review paper entitled “Reprogramming the Apoptosis–Autophagy Axis in Glioblastoma: The Central Role of the Bcl-2/Beclin-1 Complex and Survival Signalling Networks” provides a comprehensive overview of the current understanding of the Bcl-2/Beclin-1 complex and its function in regulating the balance between apoptotic and autophagic processes in glioblastoma.
The manuscript is well written, clearly describing relevant molecular mechanisms and the challenges associated with translating these insights into effective therapies for glioblastoma. I recommend updating reference no. 40 with a more recent publication to ensure the cited evidence reflects the latest advances in the field. Additionally, including information about the software and tools used to generate the illustrations in Figures 2 and 3 would improve reproducibility and clarity. These suggestions do not affect the scientific significance of the paper.
Author Response
Authors’ response to the Reviewer 1
We would like to express our gratitude for reviewing our manuscript and for constructive comments and appreciation of our work.
Regarding the comments of the Reviewer we are presenting our responses as below:
Comment 1: I recommend updating reference no. 40 with a more recent publication to ensure the cited evidence reflects the latest advances in the field.
Response 1: We appreciate this valuable comment. Regarding this the authors has changed the citation to: Prerna, K.; Dubey, V.K. Beclin1-mediated interplay between autophagy and apoptosis: New understanding. Int J Biol Macromol. 2022, 15, 204, 258-273. doi: 10.1016/j.ijbiomac.2022.02.005.
Comment 2: Additionally, including information about the software and tools used to generate the illustrations in Figures 2 and 3 would improve reproducibility and clarity.
Response 2: We fully agreed with this suggestion and we made some necessary corrections in the figures 2 and 3 description by adding sentence “Molecular graphics and analyses performed with UCSF ChimeraX” and by adding citation: Meng, E.C.; Goddard, T.D.; Pettersen, E.F.; Couch, G.S.; Pearson, Z.J.; Morris, J.H.; Ferrin, T.E. UCSF ChimeraX: Tools for structure building and analysis. Protein Sci. 2023, 32(11): e4792. doi: 10.1002/pro.4792. More information is expanded in the Acknowledgment section as it is required by software policy.
Once again we would like to thank for the Reviewers’ work for improving our manuscript before publication.
Warm regards,
Adrian Zającs

Reviewer 2 Report
Comments and Suggestions for Authors< !--StartFragment -->
Major Comments:
The review offers a comprehensive and well-cited examination of the apoptosis–autophagy axis in glioblastoma, focusing on the Bcl-2:beclin-1 complex. The work might benefit from a more explicit articulation of its conceptual innovations in relation to recent high-quality reviews on autophagy-apoptosis crosstalk in GBM. Clearly delineating how this review surpasses current literature would enhance its significance.
Although the writers' previous research is undoubtedly pertinent and significant, several portions, especially those detailing the PI3K/Akt/mTOR, Ras/Raf/MEK/ERK, and PLCγ1/PKC pathways, are predominantly based on the authors' own research works. A more equitable incorporation of independent studies would enhance objectivity and expand the viewpoint.
Autophagy in glioblastoma multiforme is significantly context-dependent and may produce contradictory results. While this duality is acknowledged, the review might be enhanced by a more rigorous examination of findings that present conflicting roles of beclin-1-mediated autophagy, particularly in instances when heightened autophagy is associated with unfavorable prognosis or resistance to therapy.
The heterogeneity of GBM, encompassing molecular subtypes such as IDH status, MGMT methylation, and mesenchymal vs proneural states, is inadequately incorporated into the discourse. Elucidating the differential functionality of the Bcl-2:beclin-1 axis among GBM subtypes would augment translational significance.
The therapeutic reprogramming segment is mechanistically robust yet predominantly preclinical. The authors should elaborate on the reasons why targeting the Bcl-2:beclin-1 axis has not yet been successfully implemented in clinical practice, including issues pertaining to toxicity, blood-brain barrier penetration, and route redundancy.
The study addresses various biomarkers but fails to specify which markers are most promising for therapeutic application. A concise compilation or prioritized enumeration of biomarkers, accompanied by their limits, would enhance clinical value.
The illustrations succinctly encapsulate signaling cascades and protein interactions, however they are predominantly schematic. Incorporating a figure that consolidates signaling intensity, post-translational modifications, and therapeutic intervention points into a cohesive decision-making model will markedly improve conceptual clarity.
Minor Comments:
Numerous sentences exhibit grammatical flaws or odd phrasing, especially in the Introduction and post-translational modification sections. A meticulous linguistic revision would enhance clarity.
Certain abbreviations (e.g., PCD-I/PCD-II, PLCG1 versus PLCγ1) are inconsistently employed throughout the paper and require standardization.
Figure legends adequately delineate pathway components but could enhance the elucidation of the biological meaning of each figure for those less acquainted with the field.
A limited number of references seem incomplete or exhibit formatting inconsistencies, especially in the latter reference list. These must be meticulously examined before to publication.
The conclusion succinctly encapsulates the review but might be enhanced by specifically delineating critical unsolved concerns and future research goals concerning apoptosis-autophagy reprogramming in GBM.
< !--EndFragment -->
Author Response
Authors’ response to the Reviewer 2
We would like to express our gratitude for reviewing our manuscript and for all constructive comments and suggestions correlated to our work.
Regarding the comments of the Reviewer we are presenting our responses as below:
Major Comments:
Comment 1: The work might benefit from a more explicit articulation of its conceptual innovations in relation to recent high-quality reviews on autophagy-apoptosis crosstalk in GBM. Clearly delineating how this review surpasses current literature would enhance its significance.
Response 1: We sincerely thank the Reviewer for this insightful suggestion. We agree that explicitly positioning our review relative to existing high-quality literature strengthens its contribution and clarity.
In the revised manuscript, we have added a dedicated paragraph in the Introduction (page 2, lines 70-80) that clearly articulates the conceptual innovations of our work relative to recent reviews on autophagy-apoptosis crosstalk in glioblastoma as below:
While several recent reviews comprehensively describe autophagy-apoptosis interplay in GBM, few explicitly integrate the role of TrkB receptor signalling networks (PI3K/Akt/mTOR, Ras/Raf/MEK/ERK, and PLCγ1/PKC) as central modulators of this crosstalk. Our review highlights how dysregulation of these pathways directly influences the apoptosis-autophagy decision node. Although existing literature mostly discusses the Bcl-2 family and beclin-1 in isolation, our manuscript systematically frames this interaction as a dynamic regulatory switch at the intersection of survival signalling and therapeutic reprogramming in GBM. We also emphasize a translational viewpoint that goes beyond descriptive summaries by proposing how targeting upstream signalling hubs that modulate the Bcl-2:beclin-1 axis may sensitize GBM cells to death pathways, supported by experimental and preclinical evidence.
We believe that these revisions make the unique conceptual contribution of our review clearer and distinguish it from prior work on this topic.
Comment 2: Although the writers' previous research is undoubtedly pertinent and significant, several portions, especially those detailing the PI3K/Akt/mTOR, Ras/Raf/MEK/ERK, and PLCγ1/PKC pathways, are predominantly based on the authors' own research works. A more equitable incorporation of independent studies would enhance objectivity and expand the viewpoint.
Response 2: We would like to thank the Reviewer for this constructive comment. We agree that a balanced integration of independent studies is essential to ensure objectivity and to place our interpretations within the broader context of the field.
For corresponding this suggestion, we had added the sentence “While selected examples from our previous experimental studies are discussed where they directly inform mechanistic aspects of apoptosis-autophagy crosstalk, the signalling pathways described below are supported by a broad body of independent experimental and clinical evidence.” as a continuation of the Chapter 2 before the information about the signalling network regulated by TrkB receptor and downstream pathways. In the revised manuscript, we have systematically expanded the citation base in sections describing the PI3K/Akt/mTOR, Ras/Raf/MEK/ERK, and PLCγ1/PKC signalling pathways by incorporating additional independent studies that corroborate and extend the mechanisms discussed. These newly cited works include reports on pathway overactivation in GBM, therapy resistance, and their roles in modulating apoptosis and autophagy independently of our own investigations.
New citations:
Section 2. for PI3K/Akt/mTOR pathway - Furnari FB, et al. Malignant astrocytic glioma: genetics, biology, and paths to treatment. Genes Dev. 2007 Nov 1;21(21):2683-710. doi: 10.1101/gad.1596707, Hwang YK, Lee DH, Lee EC, Oh JS. Importance of Autophagy Regulation in Glioblastoma with Temozolomide Resistance. Cells. 2024 Aug 11;13(16):1332. doi: 10.3390/cells13161332. For Ras/Raf/MERK/ERK - McCubrey JA, Steelman LS, Chappell WH, Abrams SL, Wong EW, Chang F, Lehmann B, Terrian DM, Milella M, Tafuri A, Stivala F, Libra M, Basecke J, Evangelisti C, Martelli AM, Franklin RA. Roles of the Raf/MEK/ERK pathway in cell growth, malignant transformation and drug resistance. Biochim Biophys Acta. 2007 Aug;1773(8):1263-84. doi: 10.1016/j.bbamcr.2006.10.001, Brennan CW, et al.; TCGA Research Network. The somatic genomic landscape of glioblastoma. Cell. 2013 Oct 10;155(2):462-77. doi: 10.1016/j.cell.2013.09.034. For PLCγ1/PKC - Khoshyomn S, et al. Inhibition of phospholipase C-gamma1 activation blocks glioma cell motility and invasion of fetal rat brain aggregates. Neurosurgery. 1999 Mar;44(3):568-77; discussion 577-8. doi: 10.1097/00006123-199903000-00073, Marvi MV, et al. Role of PLCγ1 in the modulation of cell migration and cell invasion in glioblastoma. Adv Biol Regul. 2022 Jan;83:100838. doi: 10.1016/j.jbior.2021.100838.
Section 3. For beclin-1 - Pirtoli L, et al. The prognostic role of Beclin 1 protein expression in high-grade gliomas. Autophagy. 2009 Oct;5(7):930-6. doi: 10.4161/auto.5.7.9227, Huang X, et al. Reduced expression of LC3B-II and Beclin 1 in glioblastoma multiforme indicates a down-regulated autophagic capacity that relates to the progression of astrocytic tumors. J Clin Neurosci. 2010 Dec;17(12):1515-9. doi: 10.1016/j.jocn.2010.03.051, Tamrakar S, et al. Clinicopathological Significance of Autophagy-related Proteins and its Association With Genetic Alterations in Gliomas. Anticancer Res. 2019 Mar;39(3):1233-1242. doi: 10.21873/anticanres.13233
Section 4. For BH3-mimetic - Maiuri MC, et al. BH3-only proteins and BH3 mimetics induce autophagy by competitively disrupting the interaction between Beclin 1 and Bcl-2/Bcl-X(L). Autophagy. 2007 Jul-Aug;3(4):374-6. doi: 10.4161/auto.4237, Karpel-Massler G, et al. Targeting intrinsic apoptosis and other forms of cell death by BH3-mimetics in glioblastoma. Expert Opin Drug Discov. 2017 Oct;12(10):1031-1040. doi: 10.1080/17460441.2017.1356286
Comment 3: Autophagy in glioblastoma multiforme is significantly context-dependent and may produce contradictory results. While this duality is acknowledged, the review might be enhanced by a more rigorous examination of findings that present conflicting roles of beclin-1-mediated autophagy, particularly in instances when heightened autophagy is associated with unfavorable prognosis or resistance to therapy.
Response 3: Thank you for this important observation. We agree that the context-dependence of autophagy in glioblastoma, and the resulting contradictory clinical and experimental associations, deserves a more rigorous and explicit treatment, particularly for beclin-1-linked autophagy in settings where enhanced autophagy aligns with unfavorable outcome or therapy resistance.
To address this, we revised the manuscript to expand the discussion of conflicting evidence (Section 3.3 lines 335-348). We believe these additions substantially strengthen the review by presenting a clearer, evidence-based framework for understanding when beclin-1-mediated autophagy aligns with favourable versus unfavourable biology in GBM.
Comment 4: The heterogeneity of GBM, encompassing molecular subtypes such as IDH status, MGMT methylation, and mesenchymal vs proneural states, is inadequately incorporated into the discourse. Elucidating the differential functionality of the Bcl-2:beclin-1 axis among GBM subtypes would augment translational significance.
Response 4: The authors thank the reviewer for highlighting this important point. We agree that GBM heterogeneity (IDH status, MGMT promoter methylation, and transcriptional state such as mesenchymal vs proneural) is essential for translational interpretation of apoptosis-autophagy crosstalk. In the revised manuscript, we have explicitly incorporated subtype-level context in section 3.3 and by adding a Table 1 in the section 4.3.
Comment 5: The therapeutic reprogramming segment is mechanistically robust yet predominantly preclinical. The authors should elaborate on the reasons why targeting the Bcl-2:beclin-1 axis has not yet been successfully implemented in clinical practice, including issues pertaining to toxicity, blood-brain barrier penetration, and route redundancy.
Response 5: We would like to thank the Reviewer for this important translational perspective. We agree that, although the mechanistic rationale for targeting the Bcl-2:beclin-1 axis is strong, clinical implementation in GBM remains limited and requires explicit discussion of the main barriers. In the revised manuscript, we have expanded Section 4 by adding a dedicated subsection - 4.4 Barriers to clinical translation.
We believe these additions strengthen the manuscript by clearly defining why clinical implementation is challenging and by outlining realistic development paths (BBB-optimized agents, tumour-targeted delivery strategies, and subtype-/state-informed combination regimens).
Comment 6: The study addresses various biomarkers but fails to specify which markers are most promising for therapeutic application. A concise compilation or prioritized enumeration of biomarkers, accompanied by their limits, would enhance clinical value.
Response 6: Thank you for this valuable suggestion. We fully agree that clinical utility improves when biomarkers are summarized in a concise and structured way, including explicit limitations. In the revised manuscript, we have addressed this point through the improvements implemented in response to prior comments.
Specifically, we added Table 1 in Section 4.3, which provides a concise compilation of the most clinically actionable biomarker contexts and explicitly links each to their clinical relevance, implications for interpreting Bcl-2:beclin-1/autophagy-apoptosis readouts, and translational use for stratification. This table operationalizes a prioritization by foregrounding clinically decisive stratifiers (MGMT promoter methylation; IDH/G-CIMP status; proneural vs mesenchymal state) ahead of more variable downstream markers (LC3/p62/beclin-1).
In addition, the revised Section 3.3 explicitly emphasizes that the prognostic/therapeutic interpretation of autophagy markers is subtype- and treatment-dependent (IDH/G-CIMP; MGMT/TMZ exposure; mesenchymal vs proneural states), clarifying why certain markers are more robust for therapeutic decision-making than static autophagy readouts alone. Finally, Section 4.4 discusses the principal limits of autophagy/apoptosis biomarkers (flux vs static IHC, tumour heterogeneity, pathway redundancy), thereby providing the limitations framework requested by the reviewer.
Taken together, in the opinion of the authors, Table 1 and the accompanying heterogeneity/limitations text now provide a clinically oriented compilation and practical prioritization of biomarkers for therapeutic application, while clearly stating their constraints.
Comment 7: The illustrations succinctly encapsulate signaling cascades and protein interactions, however they are predominantly schematic. Incorporating a figure that consolidates signaling intensity, post-translational modifications, and therapeutic intervention points into a cohesive decision-making model will markedly improve conceptual clarity.
Response 7: We would like to express gratitude for Reviewers’ valuable suggestion and agree that an integrative decision-making model could further enhance conceptual clarity. In the current revision, however, we have chosen to retain the existing figure set (Figures 1–3) because each figure is designed to address a distinct level of ours review’s framework - upstream survival signalling architecture centred on TrkB and its downstream pathways (Figure 1), then structural domain organisation of Bcl-2 and beclin-1 (Figure 2), and finally the Bcl-2:beclin-1 interaction interface as the mechanistic checkpoint connecting apoptosis and autophagy (Figure 3). Together, these schematics provide a stepwise representation from signalling inputs to molecular interaction output, which we believe maintains clarity without adding redundancy or increasing graphical complexity. To respond to the Reviewer’s request within this structure, we strengthened the textual integration. We hope this preserves the conceptual cohesion the Reviewer seeks while keeping the figures streamlined and focused.
Minor Comments:
Comment 8: Numerous sentences exhibit grammatical flaws or odd phrasing, especially in the Introduction and post-translational modification sections. A meticulous linguistic revision would enhance clarity.
Response 8: We thank the reviewer for this helpful comment and agree that language clarity is essential for readability. In response, we performed a thorough linguistic revision of the manuscript, with particular attention to the Introduction and the post-translational modification section. We corrected grammatical errors, improved sentence structure, standardized terminology and tense, and revised awkward phrasing to enhance clarity and precision while preserving the intended scientific meaning. We believe these edits substantially improve the overall readability of the manuscript.
Comment 9: Certain abbreviations (e.g., PCD-I/PCD-II, PLCG1 versus PLCγ1) are inconsistently employed throughout the paper and require standardization.
Response 9: Thank you for this important observation and agree that consistent abbreviation usage is essential for clarity. In the revised manuscript, we have systematically standardized abbreviations throughout the text, including consistent use of PCD-I (apoptosis) and PCD-II (autophagy), and we ensured that each abbreviation is defined at first mention and used uniformly thereafter. Regarding PLCG1 vs PLCγ1, we also clarified terminology to avoid ambiguity: PLCG1 refers to the gene, whereas PLCγ1 denotes the protein product encoded by this gene. We have now applied this convention consistently across the manuscript (including the main text, figure legends, and the abbreviation list), using PLCG1 when discussing gene expression/transcriptomic analyses and PLCγ1 when referring to the protein, its activation, or signaling function.
Comment 10: Figure legends adequately delineate pathway components but could enhance the elucidation of the biological meaning of each figure for those less acquainted with the field.
Response 10: The authors would like to thank the Reviewer for this helpful suggestion and agree that figure legends should guide readers beyond listing components, particularly for non-specialists. In the revised manuscript, we have expanded the legends for Figures 1-3 to include brief, plain-language statements describing the biological meaning and take-home message of each figure. We believe these additions improve accessibility and conceptual clarity while preserving the figures’ schematic focus.
Comment 11: A limited number of references seem incomplete or exhibit formatting inconsistencies, especially in the latter reference list. These must be meticulously examined before to publication.
Response 11: Thank you for this noticed. We have re-checked the entire bibliography, with particular attention to the latter portion of the reference list, and corrected incomplete entries (e.g., missing authors, year, volume/pages, or DOI) and any formatting inconsistencies.
For reference management, we use Mendeley, and we have now re-exported the reference list using the journal-appropriate citation style to ensure uniform formatting. We will also perform a final verification during the production stage to confirm that all references are complete and correctly formatted according to the journal requirements.
Comment 12: The conclusion succinctly encapsulates the review but might be enhanced by specifically delineating critical unsolved concerns and future research goals concerning apoptosis-autophagy reprogramming in GBM.
Response 12: We thank the Reviewer for this suggestion. The authors agree that explicitly stating unresolved questions and future directions would strengthen the translational impact of the conclusion. In the revised manuscript, we have expanded the Conclusion to include a concise set of key outstanding challenges and priority research goals for apoptosis-autophagy reprogramming in GBM.
Once again we would like to thank for the Reviewers’ work for improving our manuscript before publication.
Warm regards,
Adrian Zającs

Round 2
Reviewer 2 Report
Comments and Suggestions for Authors
The author improve well.